# Weigh(t)ing the dimensions of social vulnerability based on a regression analysis of disaster damages

Vincent David Corvin Heß<sup>1</sup>

<sup>1</sup>Wegener Center for Climate and Global Change, University of Graz, Graz, Austria Correspondence to: Vincent David Corvin Heß (vincent.hess@uni-graz.at)

Abstract. Social vulnerability determines how natural hazards can turn into social disasters. However, large uncertainties remain on how to quantify social vulnerability in a compact and comprehensive way. A principle component analysis (PCA) combines many social indicators, such as population density or the level of education, into a single vulnerability index. Whether the individual indicators increase or decrease vulnerability is usually based on consequential reasoning or derived from previous

- studies. These assumptions are rarely tested for their applicability to the study area. In a case study for the Austrian federal 5 state of Styria we observe no correlation between disaster damages and an initial vulnerability index which was based on literature-derived assumptions about the influence of each indicator and following best practices for the PCA. We show that a regression analysis of past damages can improve the interpretation of social indicators by better representing the local situation. With the results from the regression-based analysis we can improve the PCA approach and only the updated vulnerability
- index correlates with observed damages. This indicates that the use of a PCA-based vulnerability index requires a thorough 10 understanding about the regionally specific influence of each indicator on social vulnerability to explain differences in disaster damages.

### 1 Introduction

Current risk assessments recognize the importance of vulnerability as one of the key determining factors for natural disaster risk. The fifth assessment report of the International Panel on Climate Change (IPCC) defines vulnerability as the predisposition 15 or propensity to be adversely affected (Agard et al., 2014). This broad definition leaves space for multidimensional concepts of vulnerability and includes the impairment of human beings, their livelihoods, and assets (Olsson et al., 2014). The disaster risk community originally defined vulnerability as the physical susceptibility of exposed assets to sustain damages (Thomalla et al., 2006). Since then, the social dimension of vulnerability has experienced increased attention, not only in the disaster risk

community, but especially in the climate change adaptation (Birkmann et al., 2013) and risk management literature (Fuchs 20 et al., 2012). Although social vulnerability became an essential part of vulnerability research, large conceptual differences of social vulnerability remain.

Social vulnerability is seen as a multidimensional phenomenon encompassing a wide variety of concepts and approaches. Various attempts have been proposed to define and assess social vulnerability. It is generally accepted that it is rooted in and caused by a lack of adapting and coping capacity, a lack of resilience and/or a higher susceptibility towards natural hazards

25

(Cardona et al., 2012; Birkmann et al., 2013). This can manifest on different scales, from individuals and households to communities, municipalities and whole nations (Tapsell et al., 2010). This multi-dimensionality of social vulnerability, a lack of comparable data across regions and imprecise definitions of the term social vulnerability prevent a direct measurement of social vulnerability. We can only approximate social vulnerability through the use of proxy variables. A popular approach is the use of social indicators to capture different aspects of vulnerability and combine them into a single vulnerability index.

The vulnerability index approach calculates vulnerability scores from social indicators, such as the share of elderly citizens, the population density or the level of education. The relationship between the indicators and vulnerability is typically derived from previous studies or assumed based on logical reasoning (Cutter et al., 2003). There are mainly two different ways to select and combine indicators. In a deductive approach, a small set of well-defined indicators are selected, which should represent the main dimensions of social vulnerability. In an inductive approach, one uses a factor analysis to calculate the statistical

relationship between a large number of social indicators and select and weight the most important ones (Tapsell et al., 2010). Because the relationship between indicators and vulnerability is usually unknown, a large amount of studies focus on the inductive approach (e.g., Cutter et al., 2003, 2008; Armas and Gavris, 2013; Fekete, 2009; Khan, 2012; Koks et al., 2015; Rogelis et al., 2016; Rygel et al., 2006; Yoon, 2012; Willis and Fitton, 2016). Within the inductive approach, a principle

- component analysis (PCA) is the most common factor analysis technique. The composition of a PCA index includes certain design choices. Several studies have conducted sensitivity analysis to estimate which step in the construction of a social vulnerability index based on PCA is associated with the largest uncertainties (Schmidtlein et al., 2008; Tate, 2012). Although every step of the composition influences the results and needs to be justified, the weighting scheme of the individual indicators and the principle components cause the largest deviations.
- In this paper we construct a PCA-based social vulnerability index on a municipality level for the Austrian federal state of Styria. Initially, we assume an equal weight of each indicator and derive the sign of the influence from previous studies. The access to a large database of per capita damages on municipality basis allows us to alternatively retrieve indicator weights through the means of a regression analysis. For each municipality we regress the social indicators on per capita damages to represent the place-specific circumstances. While previous studies used regression analyses to externally validate vulnerability scores, to the author's knowledge, this is the first time it is used to estimate individual indicator weights directly. We compare the results with the PCA index and subsequently use the regression coefficients to calculate an updated PCA index. This allows us to explain a large share of the variability in disaster damages.

We structure the paper as follows: Section 2 describes the design choices for our calculation of a vulnerability index based on a PCA. As an alternative vulnerability index, we present a regression analysis of social indicators against natural disaster damages to calculate a regression analysis index (RA index). Both indices are applied in a case study. Section 3 provides an overview of the data used in our analysis. First, we describe the study area of the federal state of Styria consisting of 495 municipalities. Following is a description of all the available social indicators and their assumed influence on vulnerability. Lastly, we describe the damage database. Section 4 compares the results of both vulnerability indices and presents an updated PCA index that incorporates the results from the regression analysis. It also describes the sensitivity of the regression results

35 in relation to different hazard types. Section 5 discusses the results in comparison to other studies of social vulnerability and

details the limitations and advantages of our approach. Section 6 concludes that a PCA approach can only deliver meaningful results if we know the place-specific influence of all indicators.

#### 2 Methods: Quantifying social vulnerability

We calculate two different social vulnerability indices. First, we conduct a PCA to calculate a PCA vulnerability index with 5 equal component weights based on prevalent literature. Second, we perform a regression analysis and use the coefficients as indicator weights for the calculation of a RA vulnerability index with explicit weighting.

#### 2.1 Classic Approach: The PCA index

A literature review revealed that most studies which quantify social vulnerability on the basis of social indicators use a PCA to aggregate indicators into a social vulnerability index. A PCA reduces the dimensionality of the underlying data by transforming

- 10 the input data into a set of orthogonal vectors (the principal components). Each of the original indicators loads onto the principal components with a different weight. There are different possibilities to determine the number of components to retain for the analysis. Most studies use the Kaiser criterion, i.e. the number of eigenvalues of the covariance matrix that are greater than one (Kaiser, 1958). An improved criterion is based on a parallel analysis of a randomized matrix with the same dimensionality as the original data. To remove spurious components, principle components whose eigenvalues are smaller than that of the random
- 15 components are removed (Franklin et al., 1995; Tate, 2012; Ledesma and Valero-Mora, 2007). In our case, both criteria advise to keep three principal components. For the further analysis one needs to estimate the influence of the principal components on vulnerability, based on which original indicators load on the principle components. We use a varimax rotation to achieve higher loadings on individual components, which helps to interpret the resulting components (Kaiser, 1958). Based on the loadings, we assign each component a sign that reflects if it decreases or increases vulnerability.
- The PCA vulnerability index is then calculated as the weighted sum of the scores of the rotated principal components for every municipality. This is the most crucial step in the calculation of a vulnerability index (Schmidtlein et al., 2008; Tate, 2013). Proposed weighting schemes include equal weighting (Cutter et al., 2003), expert judgments (Vincent, 2004), explainable variance weighted sum (Schmidtlein et al., 2008) or pareto ranking (Rygel et al., 2006). Since each method gives different weight to the principle components, the resulting vulnerability scores can differ from each other. As there is no justification to use one approach over the others, we opted for the simplest and most widely used, equal weighting.

# 2.2 Regression Analysis: The RA index

Although the PCA index is based on the statistical relationship between social indicators, it still requires a priori knowledge about the sign of the influence of individual indicators on vulnerability to interpret the resulting components. Typically, the assumed influence is derived from other studies and study areas, based on expert judgments or based on logical reasoning. As

30 vulnerability is a dynamic notion, it is important to assess any indicator-based approach within the political, environmental, and socioeconomic landscape that it is being applied (Willis and Fitton, 2016). To weight the indicators appropriately, we conduct

10

25

a regression analysis of each social indicator against per capita damages to infer their influence on social vulnerability in our study area. We derive the per capita damages from the Austrian disaster fund, which covers damage from natural hazards and is described in further details in Sect. 3.3.

Rather than analyzing the intrinsic characteristics of the social indicator variance, we relate the social indicators directly to damages per capita. This ensures a direct measurement of an individual indicator's influence on social vulnerability. The aim of the regression analysis is to see which indicators have a statistically significant influence and if they increase or decrease vulnerability.

Initially, a multiple linear regression against per capita damages is done according to equation (1), where damage per capita  $(DPC_i)$  of municipality *i* is dependent on the standardized regression coefficients ( $\beta$ ) times the *P* social indicators (*SI*) and the error term  $\epsilon$ .

$$DPC_i = \beta_0 + \sum_{p=1}^{P} (\beta_{ip} \cdot SI_{ip}) + \epsilon_i$$
(1)

This simple regression does not take into account differences in disaster intensities and exposure of the individual municipalities. The resulting coefficients can be strongly biased and overestimate the importance of individual indicators. To account for differences in the number of affected households per natural disaster, we introduce the variable A. A is a proxy variable for

- 15 the disaster intensity, implemented as the number of disaster fund applications for a damage refund. A high number of applications can indicate either large-scale events, which affected multiple households simultaneously, municipalities that have been affected multiple times or both. Since the social indicators and the damage data are available for the years 2011 to 2014, we can calculate a fixed-effects model, where we include the individual years as dummy variables. This captures all the differences between municipalities that are not included in the social indicators and that do not change over time. Most importantly, this
- 20 includes differences in exposure, which we expect to play a major role but we do not expect to have changed over the short period of four years. The complete regression formula is

$$DPC_i = \beta_0 + \beta_1 A_i + \beta_2 C(year) + \sum_{p=1}^{P} (\beta_{ip} \cdot SI_{ip}) + \epsilon_i,$$
(2)

where C() denotes the categorized fixed effect dummy variables for the years 2011 to 2014. The resulting standardized  $\beta$  coefficients are multiplied by the social indicators per municipality and summed up to derive a single RA score. The RA vulnerability index is a vector of all individual municipality scores.

As an additional robustness check we split the dependent variable into different hazard types. The available categories are floods, landslides, storms and avalanches. The goal is to analyze which social indicators are hazard dependent and which are hazard independent and if it is justifiable to aggregate over all hazard types.

### **3** Data description

This section first describes the study area, for which the social indicators and the damage database were available. Following is a description of the eleven social indicators used in this study and their assumed influence on social vulnerability. This is the basis for the PCA approach. We then describe the damage database we used for the regression analysis.

#### 5 3.1 Study area

We conduct this study on the municipality level for the Austrian federal state of Styria. The analysis includes 495 municipalities ranging from small and rural Alpine settlements with less than 200 inhabitants to the capital city of Graz with about 269,000 inhabitants (as of 2014). Because Styria's landscape ranges from Alpine foothills to the steep topographic reliefs of the Alps, it is exposed to multiple natural hazards. Fuchs et al. (2015) analyzed the exposure of Styrian buildings. 13.2% (50,419) of all

buildings are exposed to natural hazards. About half of these buildings are exposed to river floods and half of them to flash 10 floods and torrential flows. Only 0.1% (460) of the buildings are exposed to avalanches, while 0.9% (3,712) buildings are exposed to more than one hazard.

### 3.2 Social Indicators

The selection of social indicators determines how well a social vulnerability index can describe the actual social vulnerability.

However, the availability of suitable social indicators on a municipality level constraints the selection. Table 1 summarizes the 15 eleven social indicators we use, their numerical range, and the expected influence on vulnerability. We collected the majority of the indicators from Statistik Austria (2016). Although eleven indicators is at the lower spectrum of number of indicators used for the creation of social vulnerability indices, they represent some of the most commonly used indicators across social vulnerability studies (e.g. Armas and Gavris, 2013; Cutter et al., 2003; Frigerio and De Amicis, 2016). With our indicators we can account for differences in municipality characteristics, the composition of different social groups and the social status.

20

### 3.2.1 Municipality Characteristics

Our first three indicators depict basic municipality characteristics. Population density potentially increases a municipality's vulnerability, since densely populated areas are more difficult to evacuate and decrease the available space for adaptive measures (Cutter et al., 2003; Cutter and Finch, 2008; Yoon, 2012). Linked to the overall population density is the average household

size, which states how many individuals share a household. Higher values are associated with larger families while lower values 25 depict a larger share of single households, the latter of which are associated with a higher vulnerability due to fewer available funds (Fekete, 2009; Rogelis et al., 2016). The number of fire station per citizens reflects a municipality's ability to respond to natural disasters. A higher ratio decreases a municipality's vulnerability (Rogelis et al., 2016).

### 3.2.2 Social Groups

We use four indicators that reflect the share of different social groups. The *share of children* (below 15 years) and the *share of senior citizens* (above 65 years) both increase the vulnerability of a municipality (Cutter et al., 2003; Koks et al., 2015; Rogelis et al., 2016). Not only are those groups less able to cope with disasters themselves, they also require the attention from other people, who then themselves lack the ability to cope with the disaster. The *share of woman* increases vulnerability, since woman have on average less available funds and are less represented in voluntary disaster relief units (Morrow and Phillips, 1999). A high *share of foreigners* is generally linked to a higher vulnerability, since they are often hindered by cultural and/or language barriers to respond to emergency situations (Cutter et al., 2003; Koks et al., 2015).

#### 3.2.3 Social Status

- The social status of municipalities inhabitants is reflected in another four social indicators. We include *per capita tax payments* as a proxy for the wealth of a municipalities inhabitants. Higher wealth allows for self-provisional protection measures and also allows to more easily respond to natural disasters (Cutter et al., 2003; Fekete, 2009). Higher wealth therefore decreases vulnerability, although the monetary damages can increase due to a higher number and value of exposed assets. On the contrary, a high *unemployment quota* is associated with a lack of funds to prepare for natural disasters and cope with them and thus is
- considered as a vulnerability increasing factor (Armaş and Gavriş, 2013). Better *education* decreases vulnerability, because higher educational levels are associated with higher income and also with a higher level of risk awareness (Cutter et al., 2003; Rogelis et al., 2016). We measure *education* as the share of people having at least secondary education. A higher *share of commuters* per municipality is associated with a higher vulnerability to natural disasters. Their occupational absence make it difficult to respond to disasters in time and they are more dependent on a functioning infrastructure (Fekete, 2009).

### 20 3.3 Damage data

For the evaluation of our vulnerability indices and for the regression analysis we worked with a damage database from the Austrian Disaster Fund. In Austria, every damage from natural disasters above  $1,000 \in$ , which is not covered by insurance, is eligible to be partially covered by the disaster fund. The dataset includes the number of applications, reported damages and compensation payments for each municipality for the years 2011 to 2014. Compensation payments for our study area were usually between 30% and 50% (average: 34% of damages). We worked with the reported losses from private damages, which include applications from households and private companies, but not communal losses. The latter are assessed separately and are not available for this study. Considering the low insurance penetration rates across Austria (Holub and Fuchs, 2009), we assume a constant share of private, uninsured to total losses across municipalities. Since we analyze relative differences between municipalities, covering only a fraction of the total costs does not change our results.

Table 1. Social indicators used for the analysis and their assumed influence on social vulnerability

| Indicator                               | Range (std.)              | Assumed influence |
|-----------------------------------------|---------------------------|-------------------|
| Population density <sup>a</sup>         | $30-2620 \ km^{-1}$ (247) | increase          |
| Average household size                  | 1.81-3.77 (0.34)          | decrease          |
| Fire stations per citizens <sup>a</sup> | 0-7.14 (1.21)             | decrease          |
| Share of children                       | 6.9–19.7 % (1.98)         | increase          |
| Share of elderly citizen                | 10.1–38.7 % (3.5)         | increase          |
| Share of women                          | 38.3–56.0 % (1.7)         | increase          |
| Share of foreigners                     | 0.0-36.1 % (2.9)          | increase          |
| Per capita tax payments                 | 651–2,883 €(232)          | decrease          |
| Unemployment rate                       | 0.0-13.2 % (1.8)          | increase          |
| Education                               | 4.7-84.3 % (4.7)          | increase          |
| Share of commuters                      | 25.5-91.9 % (10.0)        | increase          |

<sup>*a*</sup> The numerical calculations use the reciprocal values, namely population sparsity and citizens per fire stations

#### 3.4 Data Preparation

Since our goal is to conduct a PCA and a regression analysis, we need to make sure that our variables approximate a normal distribution. This is already true for most indicators except *population density* and *fire stations per citizens*, which we need to transform. For the numerical analysis, we took the reciprocal values of these originally right-skewed indicators, namely

- 5 population sparsity (area per inhabitants) and the number of citizens per fire station, which approximate a normal distribution. In the discussion of the results, however, we will continue to use the labels of *population density* and *fire stations per citizens*, as those more typical terms. The per capita damages and the number of disaster fund applications were also highly skewed. Distribution fitting revealed that they follow a heavy-tailed distribution. By taking the logarithm of the per capita damages and the disaster fund applications, we can approximate a normal distribution.
- Since the indicators are measured in different units, the PCA would give more weight to indicators with larger values and higher variance. We need to normalize the indicators to compare them scale independently. Yoon (2012) analyzes the impact of different linear transformations of social indicators for the use of a PCA and finds little importance of the chosen transformation. We choose the z-score transformation, which subtracts the mean and divides by the standard deviation. We apply this rescaling for the social indicators and the log-transformed per capita damages.

# 4 Results

In this section we first describe the results of the PCA and how we interpret the components to calculate our initial PCA index. We then describe the results of the regression analysis that lead to the RA index. We subsequently compare both indices with each other and per capita damages. Finally, we make use of the results from the regression analysis to calculate an updated PCA index that correlates well with per capita damages.

#### 4.1 Initial PCA index

Our PCA of eleven social indicators results in three principle components, which can explain 62% of the original variance. Table 2 shows the main factor loadings for each components. To derive a single vulnerability score we need to add the principle components in a way that reflects an increase in vulnerability with increasing scores. The first principle component is mainly

- influenced by *population sparsity* and *education*. Both indicators load onto it with opposite signs. The negative absolute value of the first principle component can indicate either a badly educated or a densely populated municipality, both increasing vulnerability. We interpret this component as an indicator of "urbanity". We label the second component as "age", which is influenced by the *share of children* and the *share of senior citizens*. We add the absolute value, because a high share of either group increases vulnerability. We interpret the third principle component as "wealth", because the *share of foreigners*, the *per*
- *capita tax payments*, the *unemployment rate*, and the *share of commuters*, have the highest loadings. Since all of indicators increase vulnerability and the majority load positively on "wealth", no transformation was necessary for the last principle component. Following the prevalent literature we applied no weights to the components and added them as in equation (3)

$$PCA index = -abs(PC1 \times SI) + abs(PC2 \times SI) + PC3 \times SI,$$
(3)

where PC are the varimax-rotated principle component vectors and SI is a matrix of all indicators and all municipalities.

#### 20 4.2 RA index

We conduct a regression analysis as an alternative measurement of a vulnerability score and to estimate the influence of individual social indicators on per capita damages. Instead of using the statistical relationship between indicators as represented by the principle components, we use the social indicators directly. The fixed effects regression also includes the number of disaster fund application as a proxy for disaster intensities. This approach allows us to explain 63% of the variance in per capita damages. The resulting coefficients are then used as weights for adding the social indicators into a composite RA index.

We find that only few indicators had a statistically significant (p 

Table 2. Main factor loadings of the social indicators onto the first three principle components. Interpretation is based on the indicators and transformed to reflect increasing vulnerability

| Name | Social indicator         | Factor loading | Interpretation | Variance explained |
|------|--------------------------|----------------|----------------|--------------------|
| PC1  | Population sparsity      | -0.53          | Urbanity       | 33%                |
|      | Education                | +0.49          | (-abs)         |                    |
| PC2  | Share of children        | -0.60          | Age            | 18%                |
|      | Share of elderly citizen | +0.62          | (+abs)         |                    |
| PC3  | Share of foreigners      | +0.54          | Wealth         | 12%                |
|      | Per capita tax payments  | +0.47          | (+)            |                    |
|      | Unemployment rate        | +0.47          |                |                    |
|      | Share of commuters       | -0.45          |                |                    |

and education show no significant influence (see Fig. 1). These results indicate that few of the chosen indicators actually determine the differences in social vulnerability.

Although the effect (i.e. vulnerability increasing or decreasing) is as expected for most of the indicators, we can observe notable and important differences. The most influential indicator, *population density*, is associated with lower per capita damages,

5 contrary to our expectations. The opposite is true for the *fire stations per capita*, which, to our surprise, increase vulnerability. We will discuss the reasons in Sect. 5. The results also hold true when we differentiate between hazard types. While population density and the fire stations per capita remain the most important indicators for all hazard types, the share of elderly citizens is only important for landslides. Floods are the only hazard for which the share of foreigners decrease vulnerability statistically significant and the share of commuters increase vulnerability. This is likely caused by the fact that flood damages can be avoided if people are at home or nearby. This is not true for landslides and avalanches which happen rapidly. The complete

10

regression table for the hazard-specific regression can be found in Tab. A1 in the appendix.

#### An updated PCA approach 4.3

When we correlate the RA index with the PCA index we see a small negative correlation (spearmans r = -.12, p