# Peer review of "Weigh(t)ing the dimensions of social vulnerability based on a regression analysis of disaster damages"

_Natural Hazards and Earth System Sciences, 2017_

## Referee Comment (RC1) · Anonymous Referee #1 · 11 Sep 2017

The paper contributes to the discussion on vulnerabilities to disasters by combining PCA and regression analysis, thereby improving the indicators used in the PCA. By doing this the scientific significance of the paper is considered good. However, there is a need for improving the scientific quality and presentation quality of the paper. Regarding scientific signifiance and quality, the paper should discuss the indicators used in the PCA and Regression model beyond providing short references to previous studies. True, the author engage in discussing possible reasons for why some indicators fall short of explaining vulnerabilities. A case in point is unemployment which is shown not to have any explanatory power. We know that unemployment is calculated as the share of those actively seeking work plus those having work; that is, the labour force. A

certain number of the unemployd is receiving unemployment benefits and as such not necessarily poor. Long-term unepmployed and outside the labour force could be more important. In short, the author should do some independent thinking on the subject, not just rely on previous studies. Furthermore, the author briefly states that the choice of indicators is constrained by available data. This is true, but we are left wondering whether other indicators might have been available. For instance, an unemployed person could very well have a spouse with a decent income and therefore not count as vulnerable; that is, what about data on family and households? The same goes for females. And tax payments may overlook that the most affluent citizens might be in a position to avoid tax, thereby potentially causing problems for the reliability of the data. From the paper, it seems that the regression analysis was conducted by using all the indicators (plus the added ones), without removing variables that did not show any explanatory power. I suggest the author do some more work on the reggression analysis in this respect; tahat is, one-by-one removing variables with lowest significance to see what impact this will have on the remaining variables. This also means that the reader should be told what type of regression model(s) has/have been used (e.g. step, enter) and what type of data each variable represent. Regarding the presentation quality, Table 1 states that education INCREASE the vulnerability whereas the text tells the opposite (DECREASE). More important, the variable Population density turns into Population sparsity in section 4.1 and 4.2 (also Table 2 and Figure 1). If not a typing error, this has to be explaned.

---

## Referee Comment (RC2) · Anonymous Referee #2 · 18 Sep 2017

*Summary*

The author uses regression analysis to examine the relationship between proxy indicators of social vulnerability and per capita damages in municipalities across the Austrian federal state of Styria. The author then uses the results from the regression analysis to refine a principal components analysis derived social vulnerability index. The author concludes that in order to accurately develop and interpret a social vulnerability index, the index developers need to know the "place-specific influence of all indicators". The author offers a number of useful insights in the discussion and conclusions of the paper that should have been applied in the analysis and interpretation of this work. I recom-

mend reconsidering the formation of the social vulnerability index following the advice the author gives at the end of the paper: including reporting place and domain-specific circumstances (i.e., an analysis of the relevant features of the municipalities in Styria). Additionally, the paper requires a clearly bounded operational definition of resilience that aligns with theory and the selection of vulnerability indicators. While, the methodology for analysing the data seems sound the questions and interpretations need to be reconsidered to make this a much more valuable contribution to their field. More specific comments on how to remedy this below:

*Specific Comments* Page 1, line 21: You have framed the paragraph as tension between concepts of physical and social vulnerability, but this last sentence of the paragraph indicates that there are conceptual differences within current approaches to the way social vulnerability is defined. These are two separate issues and should be clearly separated.

Page 2, line 3: I think what prevents a "direct measurement of social vulnerability" is the fact that it is a social construct and therefore cannot be directly observed no matter how precisely or universally it is defined.

Page 3, line 24: How did the author come to the conclusion that there is no justification to use one approach over the others [weighting schemes]"? Did Vincent, Schmidtlein et al., and Rygel et al. not make arguments for the benefits of their approaches? I think equal weighting is an understandable approach, but it needs to be better justified.

Page 5, line 25: The assumption that larger numbers of people in a household reduces vulnerability seem intuitive to me. This could mean that a single income is split between more people. It could indicate crowding.

Page 5, line 27-28: "fire station per citizens" is a response capacity indicator. There is no theoretical justification for why response capacity would increase or decrease exposure to physical damage from a disaster, unless fire personnel had an active role in disaster preparedness or mitigation.

[Figure]

Page 6, line 5: It would be interesting to know if women have the same degree of vulnerability in a developed European country compared to the USA where the Morrow and Phillips study (which is now almost 20 years old). It might be handy to supplement this variable selection with European based studies that have been conducted more recently.

Page 6, line 7: How are "foreigners" defined in this study?

Page 6, lines 23-24: Are the number of applications and pay-outs per capita or per household measures? As there is a significant variation in the population of the municipalities having the raw counts will cause misleading results.

Page 9, line 1-2: "These results indicate that few of the chosen indicators actually determine the differences in social vulnerability." This observation is not adequately bounded. The results neither confirm nor disprove that the variables influence social vulnerability. The results show that few of the selected proxies reduce a population's exposure to physical damage. There is no information about injuries, loss of income, the speed to some recovery proxy (e.g., people returning to households or meeting their previous level of productivity or health).

Page 9, line 4: It is difficult to interpret the population density result without knowing whether damages were adjusted for the municipality population. [This is clarified in the discussion Page 12, but should be made clearer in the methods and results sections.] Also, it is unclear if every areas had experienced a damage event within the four year assessment period. Both of these things need to be clarified before the reader can appropriately interpret the result.

Page 12, lines 18-19: The fact that social vulnerability indicators are only validated against monetary damages is a significant limitation of this paper and should be accounted for throughout the selection of the indicators to be evaluated.

Page 13, lines 9: Researchers need to develop clear operational definitions of vulnerability when developing quantitative assessments. There are generally agreed upon general theoretical definitions for disaster vulnerability, but such definitions are not designed to provide adequate conceptual boundaries for specific analyses – this is the job of the research. I suggest the author remedy this issue in their own paper in the first section.

Page 13, line 25: The author should apply this very reasonable advice in this paper. Additionally, the author should clarify the disaster phase of focus and the types of impacts being analysed (i.e., social indicators that capture the exposure to physical damage and social indicators that may exacerbate physical disaster losses before a disaster strikes.)

---

## Referee Comment (RC3) · Anonymous Referee #3 · 23 Oct 2017

This is a very nice paper, well written, and highly relevant for the advancement of social vulnerability indicator research. It is difficult finding substantial critique or, suggestions for improvement beyond what has already been mentioned by the other two reviewers. If anything, the paper is mainly on methodology, is lacking a case or example, but this in itself is also fine, since it is in focus and does provide advancement and clarity. Of course, it would be nice seeing how this transpires into significant effect when visualising vulnerability indices or, how and which individual indicators can inform differently. The only additional recommendation however, is to include a bit more critique already published on vulnerability indicators, and citing review papers. For instance, King 2001, Rufat and de Sherbinin could be added. In line 27, reasons and explanations could be added to "The citizens per fire station and the average household size decrease vulnerability for per capita damages."

Looking forward to this paper being published.

---

## Author Comment (AC1) · 1 Nov 2017

I thank referee #1 very much for the helpful and mindful comments. In the following, I respond to each point below and detail how I want to change the manuscript to consider each point. I belief the consideration of your comments will improve the manuscript significantly.

Regarding scientific significance and quality, the paper should discuss the indicators used in the PCA and Regression model beyond providing short references to previous studies. True, the author engage in discussing possible reasons for why some indicators fall short of explaining vulnerabilities. A case in point is unemployment which is shown not to have any explanatory power. We know that unemployment is calculated as the share of those actively seeking work plus those having work; that is, the labour force. certain number of the unemployd is receiving unemployment benefits and as such not necessarily poor. Long-term unepmployed and outside the labour force could be more important. In short, the author should do some independent thinking on the subject, not just rely on previous studies.

This is a very good idea. However, the description of used indicators is kept to a minimum on purpose for the following reasons: As you have shown exemplary for unemployment, the effectiveness of social indicators can be quite complex. Nevertheless, indicators are often used as-is in many PCA compositions of vulnerability indices with little or even no reference given. Without wanting to point out single papers, examples include Fekete (2009), Koks et al. (2015), Santos et al. (2017). Our purpose was to replicate this behavior and I did not engage in a deeper discussion of individual indicators in the methods section. The inaccuracy of some of our assumptions is excessively discussed in the discussion section. As you have pointed out, other, not discussed indicators are also problematic, but I do not assume it necessary to discuss every indicator to derive at my conclusions. The main conclusion is that indeed we need a deeper understanding of the social indicators to calculate a more accurate vulnerability index. This becomes clear when we compare the PCA index with the RA index, the latter one which does not depend on any such assumptions.

I suggest appending the following paragraph to section 3.2: Social indicators (Page 5, line 21) to make it clearer we are following standard practice for calculating a PCA based vulnerability index.

"We describe the composition of the indicators and their assumed influence on social vulnerability. We base our assumptions mainly on previous studies or, if an indicator

has not been used in the literature before, on logical argumentation. We compare this commonly used approach for calculating a social vulnerability index with our vulnerability index approach based on a linear regression, the latter of which requires no assumption on the influence of social indicators."

> From the paper, it seems that the regression analysis was conducted by using all the indicators (plus the added ones), without removing variables that did not show any explanatory power. I suggest the author do some more work on the reggression analysis in this respect; tahat is, one-by-one removing variables with lowest significance to see what impact this will have on the remaining variables. This also means that the reader should be told what type of regression model(s) has/have been used (e.g. step, enter) and what type of data each variable represent.

As you have correctly assumed, I initially did the regression with all indicators at the same time. I also used the step-method and removed indicators stepwise, beginning with the one associated with the lowest significance. However, we wanted to analyze which indicators are significant at all and if they increase or decrease vulnerability and if this is in line with the assumptions we made. From my point of view, for this analysis it is only important which indicators are statistically significant and which are not. So while the regression coefficients changed and the overall model quality in terms of $R^2$ increased, we derived at the same significant indicators.

This might be an interesting result, but stepwise regression is heavily discussed and sometimes even considered as data dredging. And since our main results did not change when removing variables without explanatory power the additional, but controversial, step of removing statistically insignificant variables was left out of the paper for the sake of brevity.

If you think it is crucial to include this information in an updated manuscript, I will be

happy to comply and provide a more detailed description of the regression analysis.

> Regarding the presentation quality, Table 1 states that education IN-CREASE the vulnerability whereas the text tells the opposite (DECREASE). More important, the variable Population density turns into Population sparsity in section 4.1 and 4.2 (also Table 2 and Figure 1). If not a typing error, this has to be explaned.

Thank you for spotting this mistake. I will make this clearer in the updated manuscript, but section 3.4 and the footnotes of Table 1 already state that I changed population density and fire stations per capita to population sparsity and number of citizens per fire stations for the numerical analysis and thus the representation and the analysis of the results. However, section 4.2. does indeed mix the labels, which I will correct to only use population sparsity in the methods and results section and continue to use population density in the rest of the paper. I will also correct Table 1 to indicate a decreasing vulnerability with increasing education.

---

## Author Comment (AC2) · 1 Nov 2017

Thank you very much for your valuable and helpful comments regarding my research article. Your comments provide many helpful insights on where to improve my manuscript to increase its scientific value. I will address each point below and how I plan to improve an updated manuscript to incorporate your suggestions.

*Summary*

The author uses regression analysis to examine the relationship between proxy indicators of social vulnerability and per capita damages in municipal-

ities across the Austrian federal state of Styria. The author then uses the results from the regression analysis to refine a principal components analysis derived social vulnerability index. The author concludes that in order to accurately develop and interpret a social vulnerability index, the index developers need to know the "place-specific influence of all indicators". The author offers a number of useful insights in the discussion and conclusions of the paper that should have been applied in the analysis and interpretation of this work. I recommend reconsidering the formation of the social vulnerability index following the advice the author gives at the end of the paper: including reporting place and domain-specific circumstances (i.e., an analysis of the relevant features of the municipalities in Styria). Additionally, the paper requires a clearly bounded operational definition of resilience that aligns with theory and the selection of vulnerability indicators. While, the methodology for analysing the data seems sound the questions and interpretations need to be reconsidered to make this a much more valuable contribution to their field.

Your main critique stems from a misunderstanding about the research goal of this paper. The goal was not to conduct a social vulnerability index for Styria as adequate as possible. Instead it should compare how common practices to calculate such an index can compare against an index based on real damage data. To enable this comparison we need to use a somewhat limited definition of social vulnerability and resilience that specifically addresses damages.

It is clear from your and the other referees comments that this goal is not clear enough defined in the current manuscript and a revised manuscript must clarify this goal upfront and throughout the manuscript. I will also improve the definition of social vulnerability in the introduction to achieve a clearly bound operational definition.

Page 2, line 3: I think what prevents a "direct measurement of social vulnerability" is the fact that it is a social construct and therefore cannot be directly observed no matter how precisely or universally it is defined.

I agree that this is an additional source of uncertainty and would suggest to include the following sentence at the end of the introductory discussion of vulnerability (page 1, line 23f):

"Ultimately, social vulnerability remains a social construct and therefore we cannot observe it directly, but only approximate it with proxy data."

> Page 3, line 24: How did the author come to the conclusion that there is no justification to use one approach over the others [weighting schemes]"? Did Vincent, Schmidtlein et al., and Rygel et al. not make arguments for the benefits of their approaches? I think equal weighting is an understandable approach, but it needs to be better justified.

That is a good point. While each author justifies his or her decision in some way or the other (sometimes just to compare it to equal weighting), no approach has been shown to be superior to the others. Again, this is in part because of the ambiguous definitions of vulnerability. However, not all of the algorithms are suitable for our later comparison with the RA index. In an updated manuscript I suggest to replace the sentence:

"As there is no justification to use one approach over the others, we opted for the simplest and most widely used algorithm, equal weighting."

with the following:

"Since not all of these approaches allow us to compare a PCA based with a regression analysis based vulnerability index, we opted for the simplest and most widely used algorithm, equal weighting."

Page 5, line 25: The assumption that larger numbers of people in a household reduces vulnerability seem intuitive to me. This could mean that a single income is split between more people. It could indicate crowding.

Am I guessing correctly, that you mean counter-intuitive? While crowding could indeed increase vulnerability, we followed the more common assumption of a increasing vulnerability with single person households. While this is only partially covered by the average household size, I consider this to be more important than crowding. In an updated manuscript I will, however, add the information that it could also indicate the opposite (e.g. Cutter et al. (2003)).

Page 5, line 27-28: "fire station per citizens" is a response capacity indicator. There is no theoretical justification for why response capacity would increase or decrease exposure to physical damage from a disaster, unless fire personnel had an active role in disaster preparedness or mitigation.

From my point of view, fire brigades can indeed have an active role in disaster preparedness and mitigation, e.g. by filling sand bags in time before the flood, evacuating people and assets from the ground floor and cellars, pumping out oil tanks, etc. All this can reduce experienced damages. I will clarify this in an updated manuscript version.

Page 6, line 5: It would be interesting to know if women have the same degree of vulnerability in a developed European country compared to the USA where the Morrow and Phillips study (which is now almost 20 years old). It might be handy to supplement this variable selection with European based studies that have been conducted more recently.

Thank you for this advice. I will include more recent studies from a European context that assume higher vulnerability for female population. (e.g. Fekete (2009), Holand et al. (2011))

Page 6, line 7: How are "foreigners" defined in this study?

Foreigners are defined as citizens without Austrian citizenship, I will add this information in the updated manuscript.

Page 6, lines 23-24: Are the number of applications and pay-outs per capita or per household measures? As there is a significant variation in the population of the municipalities having the raw counts will cause misleading results.

This is an important point, which we extensively discussed when we designed the research. In my dataset, the data is available per household. Damages were converted to damage per capita to assess the effect on the municipality as a whole, since the social indicators are also on a municipality level. Applications were used as a proxy variable for disaster intensity and therefore not converted to per capita applications. It should reflect the burden on a municipality, regardless if it is caused by stronger disaster events or by higher exposure (in larger municipalities).

Page 9, line 1-2: "These results indicate that few of the chosen indicators actually determine the differences in social vulnerability." This observation is not adequately bounded. The results neither confirm nor disprove that the variables influence social vulnerability. The results show that few of the selected proxies reduce a population's exposure to physical damage. There is no information about injuries, loss of income, the speed to some recovery proxy (e.g., people returning to households or meeting their previous level of productivity or health).

This is absolutely true in the current version of the manuscript. This issue will be partially solved by a more clear definition of social vulnerability (see comment to summary). I will also rephrase the cited sentence to

"These results indicate that few of the chosen indicators actually determine the *local* differences in social vulnerability *in Austrian municipalities*".

Please also note that the phrase focuses on the *differences* and not the overall influence. This is also discussed in Section 5, page 12, lines 23-31. But I will keep it in mind to be emphasize it in an updated manuscript.

> Page 9, line 4: It is difficult to interpret the population density result without knowing whether damages were adjusted for the municipality population. [This is clarified in the discussion Page 12, but should be made clearer in the methods and results sections.] Also, it is unclear if every areas had experienced a damage event within the four year assessment period. Both of these things need to be clarified before the reader can appropriately interpret the result.

Thanks you for this advice. I am not sure if I understand the first part correctly: Since the manuscript talks about per capita damages they must be adjusted to the municipality population. I will include the following sentence in Section 3.3: Damage data on page 6, line 24 to clarify this issues upfront:

"We only analyzed municipalities that experienced at least one disaster over this period. Although we miss some information about municipalities which successfully mitigated any damages, municipalities which are not even subject to natural hazards would otherwise distort our results."

> Page 12, lines 18-19: The fact that social vulnerability indicators are only validated against monetary damages is a significant limitation of this paper and should be accounted for throughout the selection of the indicators to be evaluated.

I fully agree to this statement. Again, the goal of the paper was not to create an adequate vulnerability index for Styria. We wanted to show that more focus should be given to the selection of indicators. We have shown that this is true for the case of monetary damages. It is very likely that this holds true also for other definitions of vulnerability. I suggest adding the following sentence to the conclusion to state this more clearly (Page 13, line 27):

"We have shown that when we define social vulnerability as vulnerability to physical damages not all indicators of our analysis were suitable. Although this will require further research, it is likely that this holds true for other definitions of social vulnerability."

Page 13, lines 9: Researchers need to develop clear operational defini-
tions of vulner-ability when developing quantitative assessments. There are
generally agreed upon general theoretical definitions for disaster vulnera-
bility, but such definitions are not designed to provide adequate conceptual
boundaries for specific analyses – this is the job of the research. I suggest
the author remedy this issue in their own paper in the first section.

Page 13, line 25: The author should apply this very reasonable advice in
this paper. Additionally, the author should clarify the disaster phase of focus
and the types of impacts being analysed (i.e., social indicators that capture
the exposure to physical damage and social indicators that may exacerbate
physical disaster losses before a disaster strikes.)

I agree to both comments. I will address this issue with an improved definition of social vulnerability and a clearer specification of the goal of the paper (see my main answer to your summary).

---

## Author Comment (AC3) · 1 Nov 2017

Thank you very much for your helpful additional comments and remarks.

> If anything, the paper is mainly on methodology, is lacking a case or example, but this in itself is also fine, since it is in focus and does provide advancement and clarity. Of course, it would be nice seeing how this transpires into significant effect when visualising vulnerability indices or, how and which individual indicators can inform differently.

Thank you for this suggestion. We considered visualizing the vulnerability indices via

vulnerability maps, but finally decided that the results are out of scope for the conclusions of this paper.

> The only additional recommendation however, is to include a bit more critique already published on vulnerability indicators, and citing review papers. For instance, King 2001, Rufat and de Sherbinin could be added.

These are great suggestions which I will highly consider to include in an updated manuscript, especially for an in-depth rewrite of the discussion and definition of vulnerability as used in this paper (see other referee responses and the final response).

> In line 27, reasons and explanations could be added to "The citizens per fire station and the average household size decrease vulnerability for per capita damages."

Since the result is in line with the expected influence I did not further evaluate this statement. To make this clear, I will include the information that these results were expected from the theoretical justification.
* * *

---

## Author Comment (AC4) · 1 Nov 2017

I would like to thank all three referees again for their valuable, helpful and considerate comments. In this final comment I want to take up the common remarks and explain how I want to address them in an updated manuscript to make this a much more valuable contribution to the research field of social vulnerability.

What becomes evident from the reviews is that I need to refine the definition of social vulnerability as used in this paper. Some ambiguity of the social indicators stems from the fact that they can act differently depending on what dimension of social vulnerability is under investigation. Therefore, I will make it clearer from the beginning that this paper

deals with social vulnerability related to direct and tangible losses. This clear definition allows to test the adequateness of social indicators by regressing them on damage data. Although this requires further research, I will argue why I strongly believe that we can draw the same conclusions for other dimensions of social vulnerability.

Another common remark of the referees was the inadequateness of some indicators. This shows that the assumed influence of a social indicator is to some extent subjective and depends on the definition of social vulnerability. With the regression analysis I have shown that some of these assumptions were indeed not applicable for our study. This is exactly what this paper tries to show. Up to now most studies give little to no reference to their assumptions. Our goal was not to calculate a "correct" social vulnerability index for Styria, but that such a calculation must include the place and domain specific contexts. In an updated manuscript I will clarify this research goal more clearly already in the abstract and throughout the manuscript.

For my suggestions on how to incorporate the specific minor issues and helpful recommendations, please consult the respective referee responses. I would like to address both of the main issues and the smaller issues in an updated manuscript, which I believe to be greatly improved by the referees' comments.